# Opportunities and challenges for decentralised clinical trials in sub-Saharan Africa: a qualitative study

Eric I Nebie ®,[1,2,3] Hélène N Sawadogo,[3] Peter van Eeuwijk,[1,2] Aita Signorell,[1,2] Elisabeth Reus,[1,2] Juerg Utzinger,[1,2] Christian Burri[1,2]

[1]University of Basel, Basel, Switzerland
[2]Swiss Tropical and Public Health Institute, Allschwil, Switzerland
[3]Centre de Recherche en Santé de Nouna, Nouna, Burkina Faso

**Correspondence to**
Dr Eric I Nebie;
neric83@yahoo.fr

## ABSTRACT

**Introduction** Digital health has gained traction in research and development, and clinical decision support systems. The COVID-19 pandemic accelerated the adoption of decentralised clinical trials (DCTs) as a mitigation and efficiency improvement strategy. We assessed the opportunities and challenges of a digital transformation in clinical research in sub-Saharan Africa from different stakeholders' perspectives.

**Methods** A qualitative study, including 40 in-depth semi structured interviews, was conducted with investigators of three leading research institutions in sub-Saharan Africa and Switzerland, contract research organisations and sponsors managing clinical trials in sub-Saharan Africa. A thematic approach was used for the analysis.

**Results** Interviewees perceived DCTs as an opportunity for trial efficiency improvement, quality improvement and reducing the burden of people participating in clinical trials. However, to gain and maintain an optimal quality of clinical trials, a transition period is necessary to tackle contextual challenges before DCTs are being implemented. The main challenges are categorised into four themes: (1) usability and practicability of the technology; (2) paradigm shift and trial data quality; (3) ethical and regulatory hurdles and (4) contextual factors (site-specific research environment and sociocultural aspects).

**Conclusion** The transformation from a site to a patient-centric model with an increased responsibility of participants should be context adapted. The transformation requires substantial investment, training of the various stakeholders and an efficient communication. Additionally, commitment of sponsors, investigators, ethics and regulatory authorities and the buy-in of the communities are essential for this change.

## INTRODUCTION

Digital health technologies (DHTs) are prominent innovations in global health as they hold promise to improve routine healthcare and foster the implementation of new health interventions.[1] Decentralised clinical trials (DCTs) refer to the remote conduct of trial-related activities out of the research site/clinic.[2] The implementation of DCTs focuses on the use of DHT[3] and includes all remote engagement with trial participants even without the DHT. The concept

### STRENGTHS AND LIMITATIONS OF THIS STUDY

⇒ This study provides insights into opportunities and challenges of digitalisation and decentralisation of clinical trials in sub-Saharan Africa from the perspectives and experience of investigators, contract research organisations and sponsors.
⇒ The study was conducted in well-established research institutions, characterised by diversified research portfolios pertaining to several therapeutic areas and research context and experienced investigators.
⇒ The conduct of the study after the onset of the COVID-19 pandemic and the increased adoption of alternative digital and decentralised solutions in research was an asset.
⇒ This study did not include opinions of study participants who might have different perceptions than clinical trial investigators, contract research organisation and sponsors.

of DCTs is participant-centred.[2] DCTs are being developed as a promising solution for future clinical research[4] to address the productivity crisis in drug research and development with continuously increasing costs, duration, burden and complexity of clinical trials.[5 6] The first documented exclusive DCT (ie, without any participant visit to the clinic) was the 'REMOTE trial', which provided the basis for future DCTs.[7] Prior to the COVID-19 pandemic, electronic data capture (EDC), electronic health records and telemedicine were the main digitalised areas in clinical trials. Recently, the use of wearable devices for electronic patient-reported outcome (ePRO) and electronic clinical outcome assessment, as well as artificial intelligence, were gaining interest.[8] The COVID-19 pandemic forcedly shed light on the opportunities for fast-track medical product development, while ensuring continuity of trials under the pandemic's stringent restrictions.[9] DCTs have proven to be feasible and to have potential to improve participant recruitment, retention

and representability in clinical trials,[10 11] data collection[12] and efficiency.[13] However, in a recent systematic review covering studies conducted in Asia, Australia, Europe and North America, there was little consensual added value of digitalisation in clinical research.[4]

Before the COVID-19 pandemic, DCTs were not fully covered by ethics and regulatory guidelines. Guidance on electronic records and electronic signatures were first included into the US Code of Federal Regulations Title 21, Part 11 (21 CFR Part 11). The first version and the revision 2 of the International Conference on Harmonisation (ICH) Guideline E6 Good Clinical Practices (GCP)[14] mentioned the use of digital and decentralised solutions in a few instances; yet, in a rather superficial way. Hence, the adoption of DCTs was challenging. By February 2022, only two countries (Denmark and Switzerland) had digital clinical trials included in their guidelines.[15] In-depth deliberation of the topic was initiated in 2010 with a European Medicines Agency (EMA) position paper covering electronic source data and data transcribed to electronic data collection tools and was further detailed in 2023 in the Guideline on computerised systems and electronic data in clinical trials.[16] The upcoming revision 3 of the ICH GCP and WHO guidance for best practices for clinical trials under public consultation provides more details and encourages the use of DCTs to improve clinical trials efficiency.[14 17]

In sub-Saharan Africa, the transition to DCTs is at an early stage. With the outset and rapid spread of SARS-CoV-2, DCTs were expeditiously initiated in some research institutions in sub-Saharan Africa to allow continuity of ongoing clinical trials despite the unfolding pandemic. The current study aimed to address the opportunities and challenges for the transition to DCTs, placing particular emphasis on sub-Saharan Africa.

## METHODS
### Study settings
Data were collected in two leading clinical research institutions in sub-Saharan Africa; the Malaria Research and Training Centre in Mali and the Clinical Research Unit of Nanoro in Burkina Faso. Additionally, data were obtained from four departments of the University Hospital Basel (USB) in Switzerland that are conducting clinical trials. The USB is also involved in clinical trials conducted in various settings including sub-Saharan Africa.

The two African research institutions have a long-standing clinical research experience with various sponsors from academia, product development partnerships (PDPs) and the pharmaceutical industry. These centres have strong capacities in terms of infrastructures, equipment and human resources and comply with international standards.[18] The two research institutions' catchment areas include both rural and semiurban settings; they operate in research clinics, hospitals and communities.

### Study design
We pursued a qualitative study to identify and explore the efficiency drivers of clinical development from various stakeholders' perspectives.[19] Several strategies for clinical trials efficiency were identified including study design and complexity, quality approaches (presented elsewhere) and decentralisation of clinical trials. The presented work focused on the opportunities and challenges for digitalisation and decentralisation in clinical research assessed through the qualitative interviews. This topic was triggered by the escalating use of DCTs, particularly during and after the COVID-19 pandemic.

### Interviews
The in-depth semistructured interviews were conducted with clinical researchers (principal investigators and study coordinators), sponsors (project managers, project leaders, clinical development staff, quality and regulatory affairs responsible and medical officers) and contract research organisations (CROs) managers who have been involved in at least one clinical trial in the past 3 years in sub-Saharan Africa. Three different interview guidelines were used for the investigators, sponsors and CROs. The main themes covered during the interviews were the interviewees' experience with DCTs before and during the COVID-19 pandemic, their perceptions of DCTs, potential benefits for sponsors and sites and, the perceived challenges for DCTs implementation.

A scientific advisory committee, comprising representatives from each sponsor type and a sub-Saharan Africa research institution leader (one from pharmaceutical industry, for profit; one from a biotech company, for profit; one from a PDP, not for profit and one from academia), reviewed the interview guides. Three members were working in Swiss-based companies with clinical research experience in sub-Saharan Africa and one in a research institution in Burkina Faso. The committee members provided inputs based on their expertise in clinical research. They were not interviewed and did not have any direct relation with the visited sites. The study was neither commissioned nor influenced by any type of sponsor.

### Data collection and analysis
The data were collected from December 2020 to October 2021. Interviews were either performed face to face (during the 6-week stay in the sites) or virtually using a Zoom licence maintained by the Swiss Tropical and Public Health Institute (Swiss TPH; Allschwil, Switzerland). The interviews were conducted by the first author either in English or in French, according to the interviewees' preference. The interviews lasted between 45 and 90 min, depending on the interviewees' experience. The interviews were recorded, transcribed, translated, coded and analysed using MAXQDA V.20 (VERBI Software 2021; Berlin, Germany). A deductive and inductive thematic analysis was pursued. Two researchers (EIN and HNS) independently performed the coding after an extended reading of the transcripts. Subcodes and codes

were generated and grouped into themes. Some preconceived themes were identified from the existing literature as the deductive analysis. Data were inductively analysed and new themes identified.

## Researchers' characteristics, reflexivity and trustworthiness

The study was conducted by researchers of Swiss TPH and the Centre de Recherche en Santé de Nouna in Burkina Faso. All involved researchers have extensive experience in the conduct of clinical studies. In addition, all of them had received specific trainings in qualitative research and made previous contributions to mixed method studies about the quality and efficiency of clinical studies, and immunisation mainly in resource-constrained settings.[20–23] The researchers did not have working experience with the sites where the study was conducted. EIN was introduced to the teams by Swiss TPH senior staff through the site leaders. Participants were given the opportunity to ask for clarifications after reading the study information sheet. The research team was aware that a commissioned study by the sponsor could induce significant bias. Hence, it was mentioned clearly to participants that the study is conducted independently and not commissioned by any sponsor. Participants felt comfortable to share their views regarding the topic. EIN and HNS are based and worked in sub-Saharan Africa and have a deep understanding of the research context. The credibility was ensured by selecting participants with different roles and responsibilities in clinical trials. As the data were subsequently collected in different research sites, it was safeguarded that the interviews were not affected by the insights gained in the different contexts and the interviews were similar and comparable. The findings were discussed within the team to ensure that the results were correctly interpreted within the context. The current findings could be transferred in research environments sharing several commonalities in sub-Saharan Africa.

## Patient and public involvement statement

The study participants were clinical research stakeholders (investigators, CROs and sponsors); the scientific advisory committee members were involved in the review of data collection tools based on their expertise. We did not include patients in the design, conduct, reporting and dissemination plan of the study.

## RESULTS

Overall, 40 in-depth semistructured interviews were conducted. Previous experience of DCTs before and after the onset of the COVID-19 pandemic, perceived context-specific DCTs opportunities and challenges, and potential solutions were discussed during the interviews. The interviewees' characteristics are summarised in table 1.

## Status of DCTs in the visited research sites

EDC, telephone calls and home visits were the main digital and decentralised solutions used in the visited sites. All

**Table 1** Interviewees' characteristics

| Participants' profiles | No |
|---|---|
| **Gender** | |
| Male | 29 |
| Female | 11 |
| Total | 40 |
| **Country of residence** | |
| Burkina Faso | 8 |
| Germany | 2 |
| Kenya | 1 |
| Mali | 9 |
| Senegal | 1 |
| South Africa | 2 |
| Switzerland | 16* |
| USA | 1 |
| Total | 40 |
| **Main role** | |
| CRO managers | 4 |
| Principal investigators | 12 |
| Sponsor PDPs | 6 |
| Sponsors pharma | 6 |
| Study coordinators | 12 |
| Total | 40 |
| **Experience in clinical research** | |
| <5 years | 6 |
| 5–10 years | 12 |
| ≥10 years | 22 |
| Total | 40 |
| **Mode of interviews** | |
| Face to face | 21 |
| Online | 19 |
| Total | 40 |

*Respectively, eight from the sponsors and eight from the University Hospital Basel.
CRO, contract research organisation; PDPs, product development partnerships.

the ongoing studies were using electronic data collection tools. In turn, none of the interviewees in sub-Saharan African sites had so far used wearable sensor devices for remote monitoring of clinical trials participants. At community level, home visits were carried out with the support of community health workers, as evidenced by the following statement: '*Basically, we are working with fieldworkers who perform home visits when we do not want to invite the participant to the site*' (male, investigator, Burkina Faso).

Interviewees' perceptions on the potential of DCTs were mixed, and hence, should be interpreted in context. Although most of the investigators and sponsors interviewed perceived DCTs as an asset for clinical research in

sub-Saharan Africa, they stressed the need to address the challenges for their adoption and implementation.

## Opportunities for DCTs
The opportunities for DCTs implementation were grouped into four main themes.

### Site empowerment and sustained site-sponsor collaboration
DCTs offered the possibility to pursue ongoing clinical trials especially during the COVID-19 pandemic. Despite travel restrictions during the pandemic (both international and domestic travels), the visited sites did not pause their ongoing trials for long periods (less than 6 months). The monitoring visits performed remotely by the CROs during the pandemic were perceived as efficient as the conventional approach. Owing to their clinical research experience with the sponsors, the sites experienced a shift towards more delegated tasks allowing them to take more responsibilities within trials. This experience was perceived successful, as indicated by a principal investigator: '*COVID-19 has ensured that this development [DCTs] is evolving and we should explore it and move forward in this area. We have already started, and it is very successful. For example, in this study of [compound] conducted with [pharmaceutical company], that we completed—it was all virtual. […] we do not have any difference. The sponsor knows our institution already—so, for experienced centres, it remains a possibility*' (male, principal investigator, Mali).

### Improved time and cost efficiency
CROs, investigators and sponsors perceived DCTs as an optimal approach for cost optimisation and timesaving when conducting clinical trials. The increased adoption of virtual meetings and training activities during the COVID-19 pandemic reinforced the need of change in current trial operations. A senior CRO manager emphasised the time, energy and cost savings through travel reduction: '*Some of the trainings could be done virtually, which probably saves some travel, so there is definitely a cost-efficiency that can be real*' (male, CRO manager, Kenya).

However, these perceptions are controversial for some sponsors and investigators arguing the potential drawbacks of exclusive virtual engagement with teams, stressing that there is no real overall 'cost saving' since more investments and time will be needed to finalise the trial due to quality issues. Furthermore, there is a downturn of experience sharing opportunities, as mentioned by a senior sponsor representative: '*You can conduct trainings and meetings virtually, but you will lose the experience sharing. It does not seem to be less expensive because you do not have to pay for the flights, but in the end, it will not reduce the costs of the study because it will delay the reporting*' (male, sponsor staff, Switzerland).

### Trial burden
Investigators and sponsors supported that DCTs could decrease the burden of trials on participants in terms of travel and time, particularly for participants living in difficult-to-reach areas. This could improve the geographical diversity in participants' recruitment. '*It can be a way to reduce the burden of trials on patients. They don't have to come to the site, five times…*' (female, sponsor staff, Switzerland). Similarly, for the sites, EDC was perceived to alleviate the burden of data entry after the data collection.

### Trial quality, documentation improvement and efficiency
DCTs promoted hybrid or centralised monitoring as well as remote and real-time access to the data allowing for timely corrective measures. This perception was also shared by CRO managers to speed-up trial monitoring. Digitalised documentation, electronic case report forms and trial files (investigator site file and trial master file) facilitated the continuity of clinical trials during the COVID-19 pandemic. Investigators reported that three ongoing studies successfully changed their master files from paper based to electronic in the sites. '*Being able to use an electronic case report form and electronic trial master file improves efficiency. It might be a bit more expensive than a paper set-up, but it helps because you get your data in-house faster. I think there's a positive link between technology and clinical trials, and it can only get better*' (male, sponsor representative, Burkina Faso).

However, some interviewees, particularly sponsor's quality responsible, stressed that the trend towards the digitalisation of all the processes of clinical trials with the use of various electronic platforms and systems could also become burdensome and a threat to trial quality. Hence, integrated and/or interoperable systems will be needed to avoid additional workload and errors due to loss of data or operational inaccuracies: '*If you have different suppliers managing that trial [DCTs], then those different suppliers will most likely come up with their own systems [data management, randomisation, lab supplies, CTA management]. There will be several systems in a given trial […] that are not connected at all*' (male, sponsor representative, South Africa).

All interviewed sponsors perceived DCTs as an opportunity to open the discussion on innovation in clinical research, especially after the lessons learnt from trials conducted during the COVID-19 pandemic. The interviewees stressed the need of innovative tools to ensure trial quality and considered DCTs as a promising option. '*There will be lessons to be learnt, particularly about the way we conduct trials, and we may have to think about new ways of ensuring quality*' (female, sponsor representative, Switzerland).

## Main challenges to the digitalisation of trials in sub-Saharan Africa
Challenges for digital transformation in clinical trials are multifactorial. The main identified challenges were grouped under four categories and included (1) the usability and practicability of the technology; (2) paradigm shift and trial data quality; (3) ethics and regulations and (4) site contextual factors (sociocultural, site-specific challenges and participant literacy) (figure 1).

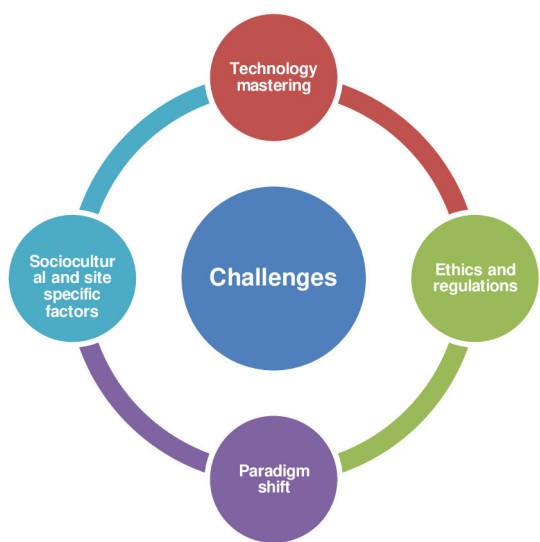

**Figure 1** Challenges for decentralised clinical trials in sub-Saharan Africa.

## Usability and practicability of the technology

The ability of the investigators and participants to master novel technologies within their daily work was raised as a fundamental challenge. Furthermore, the practical use of the technology and the lack of stable internet connectivity in some rural settings of sub-Saharan Africa were also mentioned as key challenges. The maintenance of the digital tools could be complex too, as was mentioned by some investigators: '*When you collect the data electronically—when something goes wrong, how will you take care of it? There are few things that need to happen before we can do it*' (male, study coordinator, Burkina Faso).

Moreover, the initial setup costs, the robustness, easiness of use and the acceptability (especially for wearable devices) were also mentioned as essential barriers to be address.

## Paradigm shift and trial data quality
### Shift from conventional clinical trials to DCTs

Fully DCTs will require a complete reorganisation and change of site processes, which means a substantial shift in clinical trial paradigm. '*It's true that we may be moving towards that [DCTs], but in reality there will be a lot of efforts to bring people up to speed in order to achieve that. I dare to believe that COVID-19 or other similar events will not stop the usual process that we know*' (male, principal investigator, Mali).

On the one hand, interviewed investigators stressed that the change from conventional clinical trial operations to DCTs could jeopardise the quality of clinical trials at the initiation phase and pose a threat to data validity. They suggested that changes should be limited and they would prefer the conventional or hybrid DCTs. Hence, a careful and timely planning for a consistent digital transformation is necessary. '*After COVID-19, we should go back to the old system, we should not digitalise everything*' (male, study coordinator, Mali).

On the other hand, investigators stressed the ability of sites to adjust to new technologies if there is sufficient

information about these tools. '*We have been able to collect data electronically, and we have not had any problems. We can adapt in the same way as we have adapted to electronic case report forms and others*' (male, principal investigator, Burkina Faso).

### Shift from a site centric to patient centric

Sponsors and investigators also highlighted the participant-centric nature of DCTs as a challenge, particularly in settings with modest literacy rates. '*I don't think it's impossible, but it's going to be difficult, because we're in a context where this type of study [full DCTs] involves the participant much more in the implementation of the trial, which assumes a certain level of literacy. It's going to be quite complicated at the moment*' (male, principal investigator, Burkina Faso). However, some investigators stressed that this perception could be mitigated with adequate information and instructions about the tools intended for participants use.

## Ethics and regulations

The fit of DCTs within clinical trial regulations and their acceptance by ethics committees and regulatory authorities were identified as obstacles by investigators. The discomfort resulting from the use of some technologies and the challenge for ethical clearance in the countries were the main arguments. This is reflected in the following statement regarding wearable devices: '*I think that ethically this will cause problems with the ethics committees because it's like an additional hygiene that we impose to the patients, if you have to wear a 'heart monitor' for 24 hours, plus two or three other things, it won't be easy*' (male, principal investigator, Mali).

However, some investigators suggested mitigation strategies to address this challenge. These strategies included adequate communication, education of participants and investigators, and regulatory authority bodies regarding the risk–benefit assessment of DCTs.

## Contextual factors
### Site-specific research environment

Site experience in clinical research is key for implementation of DCTs. The main research focus in the visited African institutions was malaria, which could be treated in ambulatory care when uncomplicated without a long-term follow-up. This trend of therapeutic area specificities in our study was anticipated and may be viewed as a limitation for some aspects of DCTs implementation. In this context, some aspects of DCTs (remote participants follow-up) were mentioned to be beneficial only for long-term follow-up data collection. '*Most of the measurements are done in the clinic. I don't think we can take that approach [full DCTs] with a malaria trial during the acute phase. In post-treatment, possibly, there might be a place for gathering adverse event information. Honestly, it's probably easier to have a healthcare worker or a nurse from the site visiting the community and gathering that information*' (female, sponsor representative, Switzerland).

## Sociocultural aspects

Clinical trials' social acceptability and populations' adherence were perceived as barriers to the implementation of DCTs, as indicated by the notion of a respondent: '*The main challenge for us will be how to convince the patient*' (female, study coordinator, Burkina Faso).

Furthermore, interviewees also mentioned that the tools could create misunderstanding, trust disruption and dampen population participation in future clinical trials. Adequate and timely communication was suggested to address this issue. '*I think that the socio-cultural context can also be a blocking factor. If we use these technologies, this can initially break the existing trust between researchers and populations*' (male, study coordinator, Mali).

Both sponsors and investigators also mentioned the human aspect and the risk of disruption of the relationship between investigators (as care providers) and participants. '*Physician-to-patient relationship is very important. We should not dehumanise trials as long as it is possible to do trials without putting investigators and volunteers at risk*' (male, principal investigator, Mali).

## DISCUSSION

We assessed the opportunities and challenges of DCTs in sub-Saharan Africa from the perspectives of CROs, investigators and sponsors. In light of these perspectives, we discuss potential solutions for efficient DCTs transformation. It is worth noticing that the interviewed stakeholders shared similar perspectives about DCTs in sub-Saharan Africa. Moreover, the research sites had consistent perspectives on DCTs implementation in their research context. Despite the multifactorial challenges reported by the interviewees, the benefits and potential of DCTs are important for clinical research improvement in sub-Saharan Africa. Several considerations are important for an efficient transition to DCTs in sub-Saharan Africa.

### DCTs technology mastering and context adaptation

Some aspects of the digitalisation such as the use of wearable devices for decentralised participant monitoring remain challenging. Adaptability and acceptability are important considerations in the process of transformation from conventional clinical trials to DCTs. Recently, the adoption of DCTs during the COVID-19 pandemic was essential to ensure the continuity of ongoing trials in various medical indications, initiate COVID-19-related trials[24–26] and maintain sponsor–CROs interactions.[27] The abilities of all the stakeholders including research teams, sponsors, regulatory authorities, ethics committees and participants, to master DCTs components is essential for the transition. Sponsors, sites and, where applicable, the supporting CROs should perform an early and appropriate feasibility assessment, including site capacities and context prior to conduct DCTs and suggest adaptive improvements where necessary. User-friendly and multifunctional digital solutions (with possibility to measure several parameters, for example, pulse, temperature and respiratory rate with one device) developed in sub-Saharan Africa and/or pilot testing are warranted. The multiplicity of the platforms used in DCTs for data acquisition and trial operations may become burdensome to the research teams and participants. Hence, these platforms should be carefully selected with guidance from investigators and participants who are the end users. Furthermore, the integration and interoperability of these platforms is required.[28] The availability or development of local technical expertise for the technology maintenance will be key for a long-term continuity of DCTs. Thus, the commitment of sponsors to support sites for initial investment and the continuous training will be essential.

The clinical research landscape in sub-Saharan Africa is still by far dominated by infectious diseases with most of them being acute illnesses except for tuberculosis and HIV. As the research institutions in this region are adapting to address the epidemiological transition with the rising burden of chronic and non-communicable diseases, DCTs are expected to play an important role.

### From site-centric to participant-centric approaches

Patient-centric technologies should be context adapted. In fully DCTs, participants have important responsibilities for data collection, outcome reporting and/or clinical outcome assessment. These activities were previously conducted onsite by the research teams and sometimes with the support of community health workers. The shift from site-centric to participant-centric trials could be challenging and the type of data to be 'outsourced' to the patient should be critically assessed within the context. ePROs are an important component of this change. A substantial training and clear information of trial participants and investigators and a well-defined validation process of the patient derived data is necessary. This shift also brings up the question about the compensation of research participants' time and direct costs of the technology use since they are taking over some of the tasks of the research team.[29] To mitigate acceptance issues and increase participants' buy-in, patient groups should be involved in the selection of patient-centred technologies whenever possible to enhance the success of DCTs.[30 31] When possible, DCTs should be integrated as a part of patients' follow-up with the support of their caregivers (health professionals) in the routine health facilities/hospitals. In turn, this will reinforce the research teams and caregivers collaboration. This collaboration with peripheral health facilities and hospitals of the research catchment area with clear tasks delegation can mitigate the reduced direct, face-to-face interactions with the research team. Lastly, it is important to ensure that the mentioned shift of responsibility to the patients will not waive or delay the medical care and the trial oversight by the principal investigator as well as the sponsor's responsibilities. In this light, priority should be given to transparent communication, the interaction between investigators and participants, and the participants'

ability to attend and meet the study team anytime during trial participation.

## DCTs for clinical trial efficiency and quality improvement

The ultimate goal of DCTs is to improve trial quality and efficiency (processes, costs and duration). For instance, although broadly in use, EDC still bears the challenge of source data availability for verification and data reconciliation. This gap and the lack of clear guidelines for the tools in use as source data can also elicit ethics and regulatory authority's reluctance or hesitancy for the transition to DCTs. The challenge of missing data in DCTs due to poor quality of participant recordings or lack of interoperability should be addressed.[32] The research team should ensure that the data reporting is similar and standardised for all the participants. The risk of under-reporting or excessive reporting of adverse events, for instance, will be variable across participants. This could be an important threat to the trial data validity.[29] Hence, a clear validation process and regular monitoring of the data collected are required. In the sub-Saharan Africa context, decentralised data collection with regular home visits by trial staff for participants monitoring or drug delivery combined with EDC, could be an alternative solution. Recently, hybrid DCTs approach was proven to be more efficient than fully onsite data collection.[33] Moreover, the conventional recruitment strategy with direct engagement with the community is also proven to be efficient with a low rate of loss to follow-up;[34] this should not be waved with DCTs. A hybrid recruitment model with media and online advertisement of trials on social media,[35] combined with on-site community engagement also warrants further consideration.

## Ethical and regulatory considerations to DCTs

With regard to the ethical and regulatory considerations for DCTs, one interesting aspect raised by the investigators was a potential threat that could arise from the use of the multiplicity of digital tools, particularly for patient reported outcomes or clinical outcome assessment. The extent of change to the participant daily habits and hygiene due to the use of digital technologies during a trial should be assessed and participants well informed before participation in a trial. This could potentially be a threat to participants' well-being as recently reported during an assessment of the ethical challenges and opportunities for DCTs.[29] Other critical aspects related to the electronic informed consent and participants' privacy were further mentioned.[29] This is also relevant in our research context in sub-Saharan Africa where the informed consent is often challenging.[36]

The need of sub-Saharan African ethics and regulatory authorities' guidance and adoption for the DCTs deployment is essential. The COVID-19 pandemic has fostered the deployment of DCTs and even decentralised performance of regulatory authorities functions such as inspections during the pandemic.[37] Hence, DCTs were addressed by the US Food and Drug Administration[38] and the EMA

and their implementation was encouraged. These guidelines triggered pharmaceutical sponsors' interest in DCTs, which was previously dominated by investigator-initiated trials.[39] However, in sub-Saharan Africa, as the African Medicine Agency and regional African Medicines Regulatory Harmonisation authorities were not fully operating, most of national regulatory authorities have thus far not revised their guidelines. This contributed to the gap in DCT implementation and their oversight by both ethics and regulatory authorities.[40] Investigators and sponsors could play an important role for regulatory and ethics bodies by providing them with updated information and training on available digital and decentralised solutions.[8] Furthermore, study protocols should specifically mention the nature, risk–benefit and the rationale of the use of DCTs. This will contribute to mitigate the potential breaches of clinical trials' core principles, particularly the participants' protection.[41 42]

Several important questions remain as to whom has access, control and ownership of the trial data, and the storage place of the data. These issues were partially addressed by the current ICH E6 R2 requirements empowering the sites and investigators.[14] DCTs should not waive these rights from the sites. Trustful, clear data storage and sharing policies should be adopted in countries where these are missing, in addition to the fulfilment of other ICH requirements.

## Strengths and limitations

This study aimed at filling a current knowledge gap pertaining to the opportunities and challenges of DCTs in sub-Saharan Africa. As the data were collected during the COVID-19 pandemic, the thriving of DCTs in this period allowed the participants to better appraise DCTs in their context. The study was conducted in experienced, well-equipped clinical trial sites with a diversified research portfolio. However, the interviewed professionals and visited sites did not yet have broad experience with some digital solutions used in clinical trials, which could have limited their perception of these technologies. The focus on CROs, investigators and sponsors did not allow us to cover the study participants' perspectives and experience that might be different.

## CONCLUSION

The transition to DCTs in sub-Saharan Africa has evolved in the post-COVID-19 era. DCTs facilitated the continuity of clinical trials during the pandemic and enhanced the tasks shift from the investigators to patients. If appropriately implemented, DCTs could serve as important means to improve clinical trial efficiency. However, it is essential to assess the research context including the rationale, risks and benefits and adaptability before the deployment. The commitment of ethics committees, investigators, sponsors, and regulatory authorities, and the buy-in of the communities are essential for a successful DCT adoption. There is a need for substantial technological

and logistic investment, training and communication. As DCTs are patient centric, solid and contextualised considerations with regard to good participant–investigator relationships and mutual trust should inform and guide the DCTs transition process.

**Acknowledgements** We express our deep gratitude to all the interviewees and sites (Malaria Research and Training Centre-Parasitology in Mali, the Unité de Recherche Clinique de Nanoro in Burkina Faso, the University Hospital Basel in Switzerland) and the 'African Contribution to Global Health' teams for their support. We are grateful to our scientific advisory committee members for their support.

**Contributors** EIN contributed to the study design, data collection, including interviews, data analysis, wrote the first draft and take full responsibility of the content as guarantor. CB contributed to the design, data analysis, manuscript writing and review. HNS contributed to data analysis and manuscript review. PvE provided methodological guidance and contributed to the manuscript writing and review. ER, AS and JU contributed to the manuscript writing and review.

**Funding** This study was funded by the Swiss National Science Foundation (SNSF; Bern, Switzerland) within the project 'African Contribution to Global Health' (grant number CRSII5_183577).

**Competing interests** None declared.

**Patient and public involvement** Patients and/or the public were not involved in the design, or conduct, or reporting, or dissemination plans of this research.

**Patient consent for publication** Not applicable.

**Ethics approval** This study involves human participants and was approved by Comité d'Éthique pour la Recherche en Santé (CERS) in Burkina Faso (No 2020-12-281) Comité National d'Éthique in Mali (2020/287/CE/FMOS/FAPH) Exemption (Req.2020-01364) was obtained from the Ethics Committee of Northwestern and Central Switzerland (EKZN), Switzerland. Participants gave informed consent to participate in the study before taking part.

**Provenance and peer review** Not commissioned; externally peer reviewed.

**Data availability statement** Data are available on reasonable request.

**ORCID iD**
Eric I Nebie http://orcid.org/0000-0003-1005-7898

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
