## [Reviewer comments · BMJ Open]

This paper was submitted to a another journal from BMJ but declined for publication following peer review. The authors addressed the reviewers' comments and submitted the revised paper to BMJ Open. The paper was subsequently accepted for publication at BMJ Open.

ARTICLE DETAILS

TITLE (PROVISIONAL)	Opportunities and challenges for decentralised clinical trials in sub-Saharan Africa: a qualitative study
AUTHORS	Nebie, Eric; Sawadogo, H�el�ene N.; van Eeuwijk, Peter; Signorell, Aita; Reus, Elisabeth; Utzinger, Juerg; Burri, Christian

VERSION 1 – REVIEW

REVIEWER	van Thiel, Ghislaine University Medical Centre Utrecht, Julius Centre for Health Sciences and Primary Care
REVIEW RETURNED	07-Jun-2023

GENERAL COMMENTS	This study addresses the relevant topic of the opportunities and challenges for DDCTs in Sub-Saharan Africa. It may offer some valuable insights, but in my view, the manuscript would need major revisions to be suitable for publication. I hope the comments below will be helpful. Major revisions The aim of the study remains to a certain extent unclear and ambiguous throughout the article. The abstract, introduction, methods, and discussion all mention slightly different formulations of the research aim. For example, the abstract states that the study “assessed investigators, contract research organisations, and sponsors’ perspectives on the digital transformation in clinical research in sub-Saharan Africa” (lines 34-36). By contrast, in the introduction the aim is stated as “The current study aimed to address the opportunities and challenges for an efficient transition to DDCTs, placing particular emphasis on sub-Saharan Africa” (lines 102-103). The methods finally say the main aim is to “assess the efficiency of drug development by comparing the approaches of investigator-initiated trials, PDPs, and the pharmaceutical industry in terms of study operations, costs, and duration of clinical trials” (lines 120-122). The discussion refers again to examining the opportunities and challenges from the perspective of investigators, sponsors and CROs in lines 354-356. This should be aligned more clearly. A clear definition of DDCTs is lacking in the paper. It would be helpful to have such a definition to understand what the scope of the paper is.
---

	The choice of countries is probably inspired by convenience. In my opinion it would be important to reflect on the influence of the choice of country on the results. The authors describe in the methods section that the study did not only include interviews but also direct observations and document reviews. Even though a checklist for reporting was used, I feel there is a lack of information on inclusion (sample), analysis, and reporting of the various methodological elements. I recommend to include more specific information about these aspects for each element of the study (interviews, observation, document analysis) in the manuscript. I had problems finding the outcomes of each of these methodological elements of the study in the results section of the manuscript. For example, the manuscript states that clinical trials were observed (lines 129-130), and that several trials were selected for document analysis (lines 136-137). It is however unclear whether the trials that were reviewed and observed concerned full or partial DDCTs or regular clinical trials. A more specific description of the experience of the respondents (i.e. with which type of trials/digitalized approaches), and of the trials that were observed and reviewed, would be necessary here. Additionally, it is not further specified what predefined observation guidelines were used in lines 131-132. In the results section, it was unclear to me whether the respondents talked about their experiences with DDCTs, or whether they only give their perspective/opinion on the possibilities for DDCTs (this might also relate to the aim of the study which is a bit unclear). It would also be especially valuable to emphasize the differences and similarities of experiences and perspectives between the studied countries in the results. In the discussion, several points are brought up that have not been discussed in the results section (e.g., under the subheadings 4.1, 4.2, and 4.4 there are statements on the lack of infrastructure for patient engagement, the shift of responsibilities to patients, and privacy which were not discussed in the results). In my view it would be better to instead give an (broader) interpretation of the results based on the context-specific differences between conducting DDCTs in the different countries. These are occasionally touched upon, such as in lines 446-448, but could be emphasized more. Minor comments In general, sometimes the phrasing of sentences is unclear. For example, the statements/quotes in lines 196-199, and lines 324-326, are a bit vague and unclear. A reference is missing for the claims in lines 73-76.
--	--

REVIEWER	Vanleeuw, Lieve SAMRC, Health Systems Research Unit
REVIEW RETURNED	04-Jul-2023

GENERAL COMMENTS	Dear authors, I think this is an important and well-timed study. However, several sections of the paper need more detail. Please follow the SRQR or
---

COREQ guidelines for reporting qualitative research. In addition, several parts of the discussion read like a marketing brochure for DDTs, not objective and independent research. It's also unclear who performed this research and what their links are to the sponsors and trial investigators. I sense there might be a conflict of interest.

I have left more comments on the article in pdf and have also attached a summary of the comments.

- what is the specific aim for the qualitative part of the study?
- how does the qual study i.e. perspectives inform the main aim i.e. assess effectiveness of drug development?

Page: 7

needs a section clearly describing the following:

* we need a clear description on what happens at the site e.g. is this a hospital where participants are admitted or do participants have to come in? What activities happened at the site? And how were they impacted by COVID?

* what is meant by decentralised

* what digital tools/activities are referred to in this article e.g.

training is mentioned in the results,

list the tools/activities discussed and give a clear description for each

* details on the questionnaire, which broad themes are represented in the questions?

* a clear section on researcher reflexivity: what is the researchers' situatedness within the

research and the effect that it may have on the setting and people being studied, questions being

asked, data being collected and its interpretation e.g. who are the researchers; what is their

background; what is their relationship to the study sites, sponsors, etc; how might this have

influenced the development of the questionnaire, collection of data and most importantly analysis

of data. Please check Dodgson, Joan E. "Reflexivity in qualitative research." *Journal of Human*

Lactation 35.2 (2019): 220-222.

Page: 7

Page: 8

Page: 8

reference? whose approach is being followed? e.g. Graneheim and Lundman

Page: 8

can we detail this in the table with interviewee details

Page: 8

this might be seen as problematic since it could mean that the sponsors influenced the study that

is evaluating a new way of conducting research (DDCTs) used by them. It therefore will need more

detail:

- describe the SAC in more detail: how many members, which organisation type, role, gender,

country of origin, etc. It's important for the reader to know who was involved in the development of

	the study and the tools and how this might have influenced both - describe how they reviewed the tools, what were the changes they requested, where there any parts they objected to e.g. certain questions they wanted removed, did they suggest any questions to be added, etc. It needs to be clear to what extent their involvement influenced the study and the tools.- were any of the members of the scientific advisory committee among those being interviewed? how many, who, why?- where they only involved in review of the tool or also at a later stage i.e. data analysis Page: 9 Page: 9 can this table be changed so we can see the combination of main role with country of residence and gender. Page: 9 this title doesn't capture the paragraph. The paragraph seems to describe what happens at the site, it's unclear how this is linked to previous experience. Page: 9 the different roles and their connections need more explanation, what does each of them do, where are they based and how close is their relationship to the actual study site? suggestion to visualise their roles and relationships e.g. who instructs who. It's important to see the power and other relations between the people interviewed as this might influence their perceptions and experiences. Page: 10 Page: 10 this quote doesn't really speak to fundamental feasibility Page: 10 what is an e-CRF? please write acronyms in full Page: 10 please also include the country as it is important to see the difference in perception and experience between staff at the site (in the global South) and the lead organisation (in the global North) Page: 10 Page: 10 Page: 10 are you referring to travel restrictions between countries that would have hampered those in Switzerland to visit the site in the african country. Or are you referring to travel restrictions within the african country hampering travel for participants Page: 10 Page: 10 Page: 10 needs more detail, what kind of tasks and responsibilities, how did it work before? Page: 10 this sentence is not relevant Page: 10 Page: 11 Page: 11
--	---

	Page: 11 what is mix-monitoring? do you mean hybrid? Page: 11 Page: 11 Page: 11 this is potentially quite important?!?! Does DDCT lead to poor quality? Is this one man's opinion or a general concern? Needs a lot more detail. Page: 11 how does it do that? If participants don't come to the site, how is data collected? Page: 11 Page: 11 Page: 11 suggestion to move this to the end of 3.3.4. it will fit nicer at the end as a closing thought Page: 11 how do study coordinators and CRO managers feel about that? Do they agree? I assume that study coordinators are closer to participants than investigators and sponsors (although I'm not sure because the different roles are not clear to me) Page: 13 Page: 13 the title 'contextual factors' is too vague and broad. Rather split this section into separate sections e.g. characteristics of the trial, literacy of participants, socio-cultural Page: 14 Page: 14 that's a very strong sentiment and not objective. It's the perspective of those you have interviewed, study participants might think very differently Page: 14 Page: 14 where does this come from? A reference? Or your study, in which case this must go in the results section as lack of infrastructure is not discussed in the results Page: 14 Page: 14 Page: 14 who says this? your interviewees? Page: 15 Page: 15 Page: 15 Page: 15 this is not discussed in the intro or results. Why are DDCTs seen as a shift from site-centric to participant centric? What exactly shifted? Also hard to claim this without the input from participants Page: 15 Page: 15 what is ePRO? Don't assume the reader will know these acronyms Page: 16 Page: 16 what is ICH E6 R2? Page: 16 title of these guidelines needed or explain more about these guidelines, it's unclear what they are Page: 17
--	--

	Page: 17 this reads as if it was a result of your study, please rephrase to make it clear that this is a conclusion from other studies Page: 17 this is not discussed in the results at all so it's unclear why this is discussed here Page: 17 Page: 17 Page: 17 what is this? Page: 18 Page: 18 also participants in the trial were not interviewed, hence you don't know how they experienced the shift Page: 18 this shift is not clearly discussed in the results nor in the discussion so hard to claim here as a conclusion.
--	--

VERSION 1 – AUTHOR RESPONSE

Response to Reviewer 1 comments

Dr. Ghislaine van Thiel, University Medical Centre Utrecht

Comments to the Author:

This study addresses the relevant topic of the opportunities and challenges for DDCTs in Sub-Saharan Africa. It may offer some valuable insights, but in my view, the manuscript would need major revisions to be suitable for publication. I hope the comments below will be helpful.

Major revisions

The aim of the study remains to a certain extent unclear and ambiguous throughout the article. The abstract, introduction, methods, and discussion all mention slightly different formulations of the research aim. For example, the abstract states that the study “assessed investigators, contract research organisations, and sponsors’ perspectives on the digital transformation in clinical research in sub-Saharan Africa” (lines 34-36). By contrast, in the introduction the aim is stated as “The current study aimed to address the opportunities and challenges for an efficient transition to DDCTs, placing particular emphasis on sub-Saharan Africa” (lines 102-103). The methods finally say the main aim is to “assess the efficiency of drug development by comparing the approaches of investigator-initiated trials, PDPs, and the pharmaceutical industry in terms of study operations, costs, and duration of clinical trials” (lines 120-122). The discussion refers again to examining the opportunities and challenges from the perspective of investigators, sponsors and CROs in lines 354-356. This should be aligned more clearly.

Thank you very much indeed for this important observations. The aim of the study was clarified and made consistent throughout (see revised manuscript, abstract, lines 20-22; Introduction, lines 95-96; and Methods, lines 123-126).

A clear definition of DDCTs is lacking in the paper. It would be helpful to have such a definition to understand what the scope of the paper is.

For the sake of consistency with the current literature, we changed the terminology “digitalized and decentralised clinical trials” to “decentralized clinical trials”. The two terminologies are similar from the literature and the digitalization is seen as a component of the decentralization. The respective definition is now provided (see revised manuscript, lines 60-64).

The choice of countries is probably inspired by convenience. In my opinion it would be important to reflect on the influence of the choice of country on the results.

The sites where interviews/work shadowing took place were selected from a large number of collaborating centres in numerous countries by convenience. All three centres are involved in academic as well as industrially sponsored clinical trials, both centres in Africa also with studies sponsored by Product Development Partnerships (PDPs). The choice and characteristics of the selected sites could indeed have influenced our results. However, the sites selected represents typical settings where clinical research with various sponsor types are conducted. We have now addressed this potential influence in the discussion and study strengths and limitations sections (see revised manuscript, lines 494 - 503).

The authors describe in the methods section that the study did not only include interviews but also direct observations and document reviews. Even though a checklist for reporting was used, I feel there is a lack of information on inclusion (sample), analysis, and reporting of the various methodological elements. I recommend to include more specific information about these aspects for each element of the study (interviews, observation, document analysis) in the manuscript.

The main information in this manuscript were derived from interviews. The work shadowing/observations and the document review served to have a better understanding of the research context and to triangulate the results. The methodology of the qualitative component of the main study is described and clarity further enhanced (see revised manuscript, lines 113-137).

I had problems finding the outcomes of each of these methodological elements of the study in the results section of the manuscript. For example, the manuscript states that clinical trials were observed (lines 129-130), and that several trials were selected for document analysis (lines 136-137). It is however unclear whether the trials that were reviewed and observed concerned full or partial DDCTs or regular clinical trials. A more specific description of the experience of the respondents (i.e. with which type of trials/digitalized approaches), and of the trials that were observed and reviewed, would be necessary here. Additionally, it is not further specified what predefined observation guidelines were used in lines 131-132.

In the paper, we described the methodology of the qualitative component of the mixed method design used for the overall project. For the current paper, only qualitative interviews were fully addressing the digitalization and decentralization of clinical trials. The observation and documentary reviews focused on study operations, management, quality approaches and for the context understanding. These issues have been clarified in the methods section (see revised manuscript, lines 116 – 126).

In the results section, it was unclear to me whether the respondents talked about their experiences with DDCTs, or whether they only give their perspective/opinion on the possibilities for DDCTs (this might also relate to the aim of the study, which is a bit unclear). It would also be especially valuable to emphasize the differences and similarities of experiences and perspectives between the studied countries in the results.

In the study, interviewees were first asked to share their experience with DDCTs. Secondly, we collected interviewees' perspectives on DDCTs, as they might not have experience with all digitalization and decentralization tools in their research context. There were no major difference between the countries and groups of interviewees (CROs, investigators and sponsors). We have now revised the results section and the discussion to address this comment (see revised manuscript, lines 209-220 and lines 494-503).

In the discussion, several points are brought up that have not been discussed in the results section (e.g., under the subheadings 4.1, 4.2, and 4.4 there are statements on the lack of infrastructure for patient engagement, the shift of responsibilities to patients, and privacy which were not discussed in the results). In my view it would be better to instead give an (broader) interpretation of the results

based on the context-specific differences between conducting DDCTs in the different countries. These are occasionally touched upon, such as in lines 446-448, but could be emphasized more. The current paper focuses on DCTs in selected centres in sub-Saharan Africa. We did not notice substantial differences between the two study countries, except the research catchment areas (rural and semi-urban in Burkina; rural, semi-urban and urban in Mali). The two countries have similar burden of diseases and clinical trial site organization. The discussion section was revised accordingly to focus on the contextual interpretation of the results (see revised manuscript, lines 416-438, lines 450-458 and lines 471-486).

Minor comments

In general, sometimes the phrasing of sentences is unclear. For example, the statements/quotes in lines 196-199, and lines 324-326, are a bit vague and unclear.

While revising our piece, we carefully checked each and every sentence for language, grammar and punctuation. The aforementioned statements were also revised accordingly (see revised manuscript, lines 215-216 and lines 358 - 363).

A reference is missing for the claims in lines 73-76.

We now support this claim with a relevant reference (see revised manuscript, lines 65 - 71 and reference #8).

Responses to Reviewer 2 comments

Dr. Lieve Vanleeuw, SAMRC

Comments to the Author

Dear authors

I think this is an important and well-timed study. However, several sections of the paper need more detail. Please follow the SRQR or COREQ guidelines for reporting qualitative research. In addition, several parts of the discussion read like a marketing brochure for DDTs, not objective and independent research. It's also unclear who performed this research and what their links are to the sponsors and trial investigators. I sense there might be a conflict of interest.

I have left more comments on the article in pdf and have also attached a summary of the comments. Kind regards

Thank you for the comments and review.

We compiled all the comments and questions raised by Reviewer #2 in this response letter. The manuscript was aligned to the SRQR guidelines and the checklist attached to the current submission. As mentioned in the declaration of interest, the team does not have any conflict of interest with the drug development sponsors. The study was not commissioned by any sponsor. Funding of the study is provided by the Swiss National Science Foundation (SNSF; Bern, Switzerland) and the first author pursues his PhD within the frame of this SNSF-funded study.

Methodology: needs a section clearly describing the following:

We need a clear description on what happens at the site e.g. is this a hospital where participants are admitted or do participants have to come in? What activities happened at the site? And how were they impacted by COVID?

This is now clarified in sections 2.1 (Study setting) and 2.2 (Study design) (see revised manuscript, lines 100 -111 and lines 113 - 126).

What is meant by decentralized?

As already explained in our response to Reviewer #1, a definition has been provided (see revised manuscript, lines 60-64).

What digital tools/activities are referred to in this article e.g. training is mentioned in the results, list the tools/activities discussed and give a clear description for each

The requested information is now provided in the Introduction (see revised manuscript, lines 58 - 77).

Details on the questionnaire, which broad themes are represented in the questions?

The digitalization and decentralization was a component of a much broader approach towards understanding efficiency of clinical trials. It is not possible to give all details in the manuscript due to word count restrictions. However, the broad questions are now provided in section 2.3 (Interviews) (see revised manuscript, lines 128 - 137).

A clear section on researcher reflexivity: what is the researchers' situatedness within the research and the effect that it may have on the setting and people being studied, questions being asked, data being collected and its interpretation e.g. who are the researchers; what is their background; what is their relationship to the study sites, sponsors, etc; how might this have influenced the development of the questionnaire, collection of data and most importantly analysis of data. Please check Dodgson, Joan E. "Reflexivity in qualitative research." *Journal of Human Lactation* 35.2 (2019): 220-222.

The reviewer raises an important point, which has been addressed in section 2.5 (Researchers' characteristic, reflexivity and trustworthiness) (see revised manuscript, lines 164 - 183).

What is the specific aim of the qualitative study? How does the qualitative study i.e perception inform the main aim i.e. assessment of the effectiveness of drug development?

The qualitative component of the study aimed at identifying the drivers of efficiency in clinical research from various stakeholders' perspectives and experience. From the initial literature review conducted during the conceptualization of the study, digitalization and decentralization in clinical research were mentioned in several instances as a solution to mitigate the decay of efficiency in clinical trials.

Additionally, we identified quality approaches, study operations, ethics and regulatory aspects, governance and management as other potential areas for clinical trial efficiency improvement. We explored these aspects in the sub Saharan African context. In the current paper, we focus on the aspects of digitalization and decentralization of clinical trials. The digitalization aims to facilitate the research process and ensure timely access to data, thus contribute to reduce the duration and costs (thus the efficiency). It is important to identify the scope, benefits and challenges of this growing efficiency improvement measure from the main stakeholders of clinical research. A sentence has been added to further enhance clarity of the aim of the qualitative component (see revised manuscript, lines 116 - 126).

This might be seen as problematic since it could mean that the sponsors influenced the study that is evaluating a new way of conducting research (DDCTs) used by them. It therefore will need more detail:

- describe the SAC in more detail: how many members, which organization type, role, gender, country of origin, etc. It's important for the reader to know who was involved in the development of the study and the tools and how this might have influenced both

- describe how they reviewed the tools, what were the changes they requested, where there any parts they objected to e.g. certain questions they wanted removed, did they suggest any questions to be added, etc. It needs to be clear to what extent their involvement influenced the study and the tools.

- were any of the members of the scientific advisory committee among those being interviewed? How many, who, why?

Were they only involved in review of the tool or also at the design?

Line 175: the different roles and their connections need more explanation, what does each of them do, where are they based and how close is their relationship to the actual study site? Suggestion to

visualise their roles and relationships e.g. who instructs who. It's important to see the power and other relations between the people interviewed as this might influence their perceptions and experiences. The study was not commissioned, nor influenced by any type of sponsor. The Swiss Tropical and Public Health Institute (Swiss TPH) and its network of partners have over 30 years of experience in academic and industrial drug and vaccine research and development (R&D) in numerous functions. The quality and efficacy of clinical trials have been research topics for many years (see, for example, the following publications (1-5)). The current project stems from a Swiss TPH symposium on this topic and then became a module of the project African Contribution to Global Health (AfriCon), funded by the Swiss National Science Foundation (SNSF). The AfriCon project consortium members, including senior staff and PhD students at Swiss TPH, do not have any conflict of interest with any sponsor. The four advisory committee members were selected by the research team based on clinical research experience and their expertise of the research context in sub-Saharan Africa (one from pharmaceutical industry [i.e. for profit], one from a biotech company, one from product development partnership [i.e. not for profit] and one from academia); they were not appointed or proposed by their companies or organizations. Three members were working in Swiss-based companies with clinical research experience in sub-Saharan Africa and one in a research institution in Burkina Faso. These issues have now been clarified in section 2.3 (Interviews) (see revised manuscript, lines 138 - 146). The scientific advisory committee has pure advisory role and no decisive power over the study team. They reviewed all the tools upon request per the research team (interview guide, document review guide and work shadowing tools). The inputs comprises aspects of clarity and completeness of the documents. The committee members were not interviewed and did not have any direct relation with the visited sites.

Line 145: Can we detail this in the interviewees' details?

It is difficult to disclose more detail beyond the general roles as this will be a breach to confidentiality of the identity of our interviewees. For some of the interviewees, their role is unique in their companies and this precision might disclose their identity. Hence, no further action was taken.

Type of qualitative analysis: Read Graneheim and Lundmann

Thanks for your suggestion to check this methodological paper by Graneheim & Lundmann. We have read with interest and found the piece of considerable importance for our own research. The analysis section and reflexivity were revised accordingly (see revised manuscript, lines 148 – 162 and lines 164 - 183).

3.1. Title doesn't match with the content: this title doesn't capture the paragraph. The paragraph seems to describe what happens at the site, it's unclear how this is linked to previous experience. We carefully revised the title, so that it encapsulate the main thrust of our piece (see revised manuscript, lines 209).

Include the country in the quotes

The country names have now been included in the quotes.

What is an e-CRF? Please write acronyms in full

The abbreviation "eCRF", which stands of "electronic case report form", has been introduced upon first use (see revised manuscript, lines 269 - 270).

Remove quote in line 198

The respective quote was removed in the revised version of the manuscript.

217: are you referring to travel restrictions between countries that would have hampered those in Switzerland to visit the site in the African country? Or are you referring to travel restrictions within the African country hampering travel for

In this section, we are referring to all travel restriction (international travels and travels within the countries-domestic travels) without exception during the COVID-19 pandemic. For instance, there were limitations of travel possibilities for contract research organizations, which are organization committed by the sponsors to assist and monitor the research sites for trials implementation. In some case, the continuity of trials is influenced by the outcomes of these visits. For multi-country trials, it is common to have face-to-face investigator meetings, which were not possible in some contexts during the COVID-19 pandemic due to country-specific measures and lockdowns.

217: explain the shift of tasks needs more detail, what kind of tasks and responsibilities, how did it work before?

This issue has now been addressed in the manuscript in section 3.3.2 (Paradigm shift and trial data quality) and section 4.2 (From site-centric to participant-centric approaches) (see revised manuscript, lines 312-336 and lines 416-438).

227: Not relevant

We agree with Reviewer #2, and hence, the respective part has been omitted.

230: how do study coordinators and CRO managers feel about that? Do they agree? I assume that study coordinators are closer to participants than investigators and sponsors (although I'm not sure because the different roles are not clear to me)

CROs and study coordinators perceived DCTs as an opportunity to further enhance clinical trial work.

241: Important?!?! Does DDCT lead to poor quality? Is this one man's opinion or a general concern? Needs a lot more detail?

Digitalization and decentralization in clinical trials could be a threat to trial quality. There is growing evidence that decentralization of clinical trials could lead to ethical, regulatory and quality issues due to several factors (for a detailed discussion, see: (6-8). The remote data collection with the patients having the responsibility to record their own data without direct assistance of any study team member could be challenging when it comes to the validation of the data. One has to ensure that data collection is identical and standardized across participants to ensure their adequate interpretation. This issue has now been elaborated in more detailed in section 4.3 (see revised manuscript, lines 440 - 458).

249: How does it do that? If participants don't come to the site, how is data collected?

As per definition, in DCTs, part or all the trial activities are conducted out of the clinic as opposed to the conventional clinical trials. This can be done by several means. Patients can report their own data for the trial outcome assessment, known as electronic patient reported outcome, using either a device provided by the study team or their own devices. This remote conduct of clinical trials could also be limited to patient follow-up through the reporting of adverse events. Additionally, investigational medical product (medication used for the trial) could be delivered at the participant home. This issue has now been clarified in section 4.2 (From site-centric to participant-centric approaches) (see revised manuscript, lines 416 - 438).

255 to 260: Move this to the end as closing word

The respective sentence has been moved to the end of the section, as suggested (see revised manuscript, lines 289 - 291).

3.4.4 The title 'contextual factors' is too vague and broad. Rather split this section into separate sections e.g. characteristics of the trial, literacy of participants, socio-cultural

The title was split into two sections (see revised manuscript, lines 352 and 364).

Line 353 that's a very strong sentiment and not objective. It's the perspective of those you have interviewed, study participants might think very differently

We agree that our conclusion might not reflect those of the participants. However, the focus of our study was on investigators, CROs and sponsors. Nevertheless, we toned down the statement (see revised manuscript, lines 380 - 387).

369: Reference: where does this come from? A reference? Or your study, in which case this must go in the results section as lack of infrastructure is not discussed in the results
This statement was quoted from a previous study. We deleted it because it goes beyond the scope of the current study.

374: who says this? your interviewees?

This is an interpretation of our interviewees' data.

390: this is not discussed in the intro or results. Why are DDCTs seen as a shift from site-centric to participant centric? What exactly shifted? Also hard to claim this without the input from participants
We carefully revised the results section and provide the necessary details on the shift from site centric to patient centric. This was initially mentioned under two main sections. In the revised version, the information is now included under section 3.3.2 (Paradigm shift and trial data quality) and section 4.2 (From site-centric to participant-centric approaches) (see revised manuscript, lines 329 - 336 and lines 416 - 438).

391: what is ePRO? Don't assume the reader will know these acronyms

"ePRO" is the abbreviation of "electronic patient reported outcomes", previously spelt out and defined (see revised manuscript, line 69-70).

432: what is ICH E6 R2?

ICH E6 R2 refers to the guideline for the conduct of clinical trials, i.e. "International Conference on Harmonization (ICH) Guideline E6 Good Clinical Practices", which is defined on lines 78-91.

438: title of these guidelines needed or explain more about these guidelines, it's unclear what they are
This refers to the fundamental guideline ICH E6 R2 underlying the conduct of clinical trials – we expect our audience to be familiar with this document.

4.6: Why do we discuss community engagement? This is not discussed in the results at all so it's unclear why this is discussed here

The section was deleted and integrated into the appropriate place in the manuscript (see revised manuscript, lines 434- 438).

458: Rephrase

This sentence has been deleted in the revised version.

476: Tokenization and Ai: What is this?

Deleted with the section as this was brought in by the authors not the interviewees.

488: also participants in the trial were not interviewed, hence you don't know how they experienced the shift

Our study focused on the investigators, CROs and sponsors and this is now emphasized more clearly (see revised manuscript, lines 499 - 503).

Conclusion: this shift is not clearly discussed in the results nor in the discussion so hard to claim here as a conclusion.

The task shift from sponsors to sites has been addressed in section 3.3.2 (Paradigm Shift and Trial Data Quality) (see revised manuscript, lines 311 - 336).

References

1. Vischer N, Pfeiffer C, Burri C. Improving Efficiency and Quality in Clinical Trials in Sub-Saharan Africa. *BMJ Global Health*. 2017;2(Suppl 2):A56.2-A.
2. Vischer N, Pfeiffer C, Joller A, et al. The Good Clinical Practice guideline and its interpretation - perceptions of clinical trial teams in sub-Saharan Africa. *Trop Med Int Health*. 2016;21(8):1040-8.
3. Vischer N, Pfeiffer C, Limacher M, et al. "You can save time if..."-A qualitative study on internal factors slowing down clinical trials in Sub-Saharan Africa. *PLoS One*. 2017;12(3):e0173796.
4. De Pretto-Lazarova A, Fuchs C, van Eeuwijk P, et al. Defining clinical trial quality from the perspective of resource-limited settings: A qualitative study based on interviews with investigators, sponsors, and monitors conducting clinical trials in sub-Saharan Africa. *PLoS Negl Trop Dis*. 2022;16(1):e0010121.

5. Kuepfer I, Burri C. Reflections on clinical research in sub-Saharan Africa. *Int J Parasitol.* 2009;39(9):947-54.
6. de Jong AJ, van Rijssel TI, Zuidgeest MGP, et al. Opportunities and Challenges for Decentralized Clinical Trials: European Regulators' Perspective. *Clin Pharmacol Ther.* 2022;112(2):344-52.
7. Petrini C, Mannelli C, Riva L, et al. Decentralized clinical trials (DCTs): A few ethical considerations. *Front Public Health.* 2022;10:1081150.
8. Vayena E, Blasimme A, Sugarman J. Decentralised clinical trials: ethical opportunities and challenges. *Lancet Digit Health.* 2023;5(6):e390-e4.

VERSION 2 – REVIEW

REVIEWER	van Thiel, Ghislaine University Medical Centre Utrecht, Julius Centre for Health Sciences and Primary Care
REVIEW RETURNED	17-Aug-2023

GENERAL COMMENTS	Assessment of responses Reviewer 1 comments Dr. Ghislaine van Thiel, University Medical Centre Utrecht Please find my assessment of the author's responses in red Comments to the Author: This study addresses the relevant topic of the opportunities and challenges for DDCTs in Sub-Saharan Africa. It may offer some valuable insights, but in my view, the manuscript would need major revisions to be suitable for publication. I hope the comments below will be helpful. Major revisions The aim of the study remains to a certain extent unclear and ambiguous throughout the article. The abstract, introduction, methods, and discussion all mention slightly different formulations of the research aim. For example, the abstract states that the study "assessed investigators, contract research organisations, and sponsors' perspectives on the digital transformation in clinical research in sub-Saharan Africa" (lines 34-36). By contrast, in the introduction the aim is stated as "The current study aimed to address the opportunities and challenges for an efficient transition to DDCTs, placing particular emphasis on sub-Saharan Africa" (lines 102-103). The methods finally say the main aim is to "assess the efficiency of drug development by comparing the approaches of investigator-initiated trials, PDPs, and the pharmaceutical industry in terms of study operations, costs, and duration of clinical trials" (lines 120-122). The discussion refers again to examining the opportunities and challenges from the perspective of investigators, sponsors and CROs in lines 354-356. This should be aligned more clearly. Thank you very much indeed for this important observations. The aim of the study was clarified and made consistent throughout (see revised manuscript, abstract, lines 20-22; Introduction, lines 95-96; and Methods, lines 123-126). The aim of the study is now stated more clearly throughout the article.
---

A clear definition of DDCTs is lacking in the paper. It would be helpful to have such a definition to understand what the scope of the paper is.

For the sake of consistency with the current literature, we changed the terminology “digitalized and decentralised clinical trials” to “decentralized clinical trials”. The two terminologies are similar from the literature and the digitalization is seen as a component of the decentralization. The respective definition is now provided (see revised manuscript, lines 60-64).

OK

The choice of countries is probably inspired by convenience. In my opinion it would be important to reflect on the influence of the choice of country on the results.

The sites where interviews/work shadowing took place were selected from a large number of collaborating centres in numerous countries by convenience. All three centres are involved in academic as well as industrially sponsored clinical trials, both centres in Africa also with studies sponsored by Product Development Partnerships (PDPs). The choice and characteristics of the selected sites could indeed have influenced our results.

However, the sites selected represents typical settings where clinical research with various sponsor types are conducted. We have now addressed this potential influence in the discussion and study strengths and limitations sections (see revised manuscript, lines 494 - 503).

OK

The authors describe in the methods section that the study did not only include interviews but also direct observations and document reviews. Even though a checklist for reporting was used, I feel there is a lack of information on inclusion (sample), analysis, and reporting of the various methodological elements. I recommend to include more specific information about these aspects for each element of the study (interviews, observation, document analysis) in the manuscript.

The main information in this manuscript were derived from interviews. The work shadowing/observations and the document review served to have a better understanding of the research context and to triangulate the results. The methodology of the qualitative component of the main study is described and clarity further enhanced (see revised manuscript, lines 113-137).

In my opinion, the methods section still lacks clarity and this is also still a problem for the results section (see next comment). If the observation/document analysis is used for data-triangulation, it should be clear how these methods were used and how their results are combined with the interview data. A further revision of the manuscript to resolve these issues would in my view be necessary to make the paper acceptable for publication

I had problems finding the outcomes of each of these methodological elements of the study in the results section of the manuscript. For example, the manuscript states that clinical trials were observed (lines 129-130), and that several trials were selected for document analysis (lines 136-137). It is however unclear whether the trials that were reviewed and observed concerned full or partial DDCTs or regular clinical trials. A more specific description of the experience of the respondents (i.e. with which type of trials/digitalized approaches), and of the trials that were observed and reviewed, would be necessary here.

Additionally, it is not further specified what predefined observation guidelines were used in lines 131-132.

In the paper, we described the methodology of the qualitative component of the mixed method design used for the overall project. For the current paper, only qualitative interviews were fully addressing the digitalization and decentralization of clinical trials. The observation and documentary reviews focused on study operations, management, quality approaches and for the context understanding. These issues have been clarified in the methods section (see revised manuscript, lines 116 – 126).

This author response seems to suggest that only qualitative interviews are reported in this paper. If so, the methods should be limited accordingly. If not, the elements of observation and document analysis should be further explained, and their results clearly stated. See also comment above.

In the results section, it was unclear to me whether the respondents talked about their experiences with DDCTs, or whether they only give their perspective/opinion on the possibilities for DDCTs (this might also relate to the aim of the study, which is a bit unclear). It would also be especially valuable to emphasize the differences and similarities of experiences and perspectives between the studied countries in the results.

In the study, interviewees were first asked to share their experience with DDCTs. Secondly, we collected interviewees' perspectives on DDCTs, as they might not have experience with all digitalization and decentralization tools in their research context. There were no major difference between the countries and groups of interviewees (CROs, investigators and sponsors). We have now revised the results section and the discussion to address this comment (see revised manuscript, lines 209-220 and lines 494-503).

Acceptable

In the discussion, several points are brought up that have not been discussed in the results section (e.g., under the subheadings 4.1, 4.2, and 4.4 there are statements on the lack of infrastructure for patient engagement, the shift of responsibilities to patients, and privacy which were not discussed in the results). In my view it would be better to instead give an (broader) interpretation of the results based on the context-specific differences between conducting DDCTs in the different countries. These are occasionally touched upon, such as in lines 446-448, but could be emphasized more.

The current paper focuses on DCTs in selected centres in sub-Saharan Africa. We did not notice substantial differences between the two study countries, except the research catchment areas (rural and semi-urban in Burkina; rural, semi-urban and urban in Mali). The two countries have similar burden of diseases and clinical trial site organization. The discussion section was revised accordingly to focus on the contextual interpretation of the results (see revised manuscript, lines 416-438, lines 450-458 and lines 471-486).

Acceptable, although the patient engagement section does not seem to be linked to the results section. This could be improved.

Minor comments

In general, sometimes the phrasing of sentences is unclear. For example, the statements/quotes in lines 196-199, and lines 324-326, are a bit vague and unclear.

	While revising our piece, we carefully checked each and every sentence for language, grammar and punctuation. The aforementioned statements were also revised accordingly (see revised manuscript, lines 215-216 and lines 358 - 363). OK A reference is missing for the claims in lines 73-76. We now support this claim with a relevant reference (see revised manuscript, lines 65 - 71 and reference #8). OK
--	--

VERSION 2 – AUTHOR RESPONSE

Comments to the Author:

This study addresses the relevant topic of the opportunities and challenges for DDCTs in Sub-Saharan Africa. It may offer some valuable insights, but in my view, the manuscript would need major revisions to be suitable for publication. I hope the comments below will be helpful.

Major revisions

The aim of the study remains to a certain extent unclear and ambiguous throughout the article. The abstract, introduction, methods, and discussion all mention slightly different formulations of the research aim. For example, the abstract states that the study “assessed investigators, contract research organisations, and sponsors’ perspectives on the digital transformation in clinical research in sub-Saharan Africa” (lines 34-36). By contrast, in the introduction the aim is stated as “The current study aimed to address the opportunities and challenges for an efficient transition to DDCTs, placing particular emphasis on sub-Saharan Africa” (lines 102-103). The methods finally say the main aim is to “assess the efficiency of drug development by comparing the approaches of investigator-initiated trials, PDPs, and the pharmaceutical industry in terms of study operations, costs, and duration of clinical trials” (lines 120-122). The discussion refers again to examining the opportunities and challenges from the perspective of investigators, sponsors and CROs in lines 354-356. This should be aligned more clearly.

Thank you very much indeed for this important observations. The aim of the study was clarified and made consistent throughout (see revised manuscript, abstract, lines 20-22; Introduction, lines 95-96; and Methods, lines 123-126).

The aim of the study is now stated more clearly throughout the article.

A clear definition of DDCTs is lacking in the paper. It would be helpful to have such a definition to understand what the scope of the paper is.

For the sake of consistency with the current literature, we changed the terminology “digitalized and decentralised clinical trials” to “decentralized clinical trials”. The two terminologies are similar from the literature and the digitalization is seen as a component of the decentralization. The respective definition is now provided (see revised manuscript, lines 60-64).

OK

The choice of countries is probably inspired by convenience. In my opinion it would be important to reflect on the influence of the choice of country on the results.

The sites where interviews/work shadowing took place were selected from a large number of collaborating centres in numerous countries by convenience. All three centres are involved in

academic as well as industrially sponsored clinical trials, both centres in Africa also with studies sponsored by Product Development Partnerships (PDPs). The choice and characteristics of the selected sites could indeed have influenced our results. However, the sites selected represents typical settings where clinical research with various sponsor types are conducted. We have now addressed this potential influence in the discussion and study strengths and limitations sections (see revised manuscript, lines 494 - 503).

OK

The authors describe in the methods section that the study did not only include interviews but also direct observations and document reviews. Even though a checklist for reporting was used, I feel there is a lack of information on inclusion (sample), analysis, and reporting of the various methodological elements. I recommend to include more specific information about these aspects for each element of the study (interviews, observation, document analysis) in the manuscript. The main information in this manuscript were derived from interviews. The work shadowing/observations and the document review served to have a better understanding of the research context and to triangulate the results. The methodology of the qualitative component of the main study is described and clarity further enhanced (see revised manuscript, lines 113-137).

In my opinion, the methods section still lacks clarity and this is also still a problem for the results section (see next comment). If the observation/document analysis is used for data-triangulation, it should be clear how these methods were used and how their results are combined with the interview data. A further revision of the manuscript to resolve these issues would in my view be necessary to make the paper acceptable for publication

I had problems finding the outcomes of each of these methodological elements of the study in the results section of the manuscript. For example, the manuscript states that clinical trials were observed (lines 129-130), and that several trials were selected for document analysis (lines 136-137). It is however unclear whether the trials that were reviewed and observed concerned full or partial DDCTs or regular clinical trials. A more specific description of the experience of the respondents (i.e. with which type of trials/digitalized approaches), and of the trials that were observed and reviewed, would be necessary here. Additionally, it is not further specified what predefined observation guidelines were used in lines 131-132.

In the paper, we described the methodology of the qualitative component of the mixed method design used for the overall project. For the current paper, only qualitative interviews were fully addressing the digitalization and decentralization of clinical trials. The observation and documentary reviews focused on study operations, management, quality approaches and for the context understanding. These issues have been clarified in the methods section (see revised manuscript, lines 116 – 126).

This author response seems to suggest that only qualitative interviews are reported in this paper. If so, the methods should be limited accordingly. If not, the elements of observation and document analysis should be further explained, and their results clearly stated. See also comment above.

In the results section, it was unclear to me whether the respondents talked about their experiences with DDCTs, or whether they only give their perspective/opinion on the possibilities for DDCTs (this might also relate to the aim of the study, which is a bit unclear). It would also be especially valuable to emphasize the differences and similarities of experiences and perspectives between the studied countries in the results.

In the study, interviewees were first asked to share their experience with DDCTs. Secondly, we collected interviewees' perspectives on DDCTs, as they might not have experience with all digitalization and decentralization tools in their research context. There were no major difference between the countries and groups of interviewees (CROs, investigators and sponsors). We have now

revised the results section and the discussion to address this comment (see revised manuscript, lines 209-220 and lines 494-503).

Acceptable

In the discussion, several points are brought up that have not been discussed in the results section (e.g., under the subheadings 4.1, 4.2, and 4.4 there are statements on the lack of infrastructure for patient engagement, the shift of responsibilities to patients, and privacy which were not discussed in the results). In my view it would be better to instead give an (broader) interpretation of the results based on the context-specific differences between conducting DDCTs in the different countries. These are occasionally touched upon, such as in lines 446-448, but could be emphasized more. The current paper focuses on DCTs in selected centres in sub-Saharan Africa. We did not notice substantial differences between the two study countries, except the research catchment areas (rural and semi-urban in Burkina; rural, semi-urban and urban in Mali). The two countries have similar burden of diseases and clinical trial site organization. The discussion section was revised accordingly to focus on the contextual interpretation of the results (see revised manuscript, lines 416-438, lines 450-458 and lines 471-486).

Acceptable, although the patient engagement section does not seem to be linked to the results section. This could be improved.

Minor comments

In general, sometimes the phrasing of sentences is unclear. For example, the statements/quotes in lines 196-199, and lines 324-326, are a bit vague and unclear.

While revising our piece, we carefully checked each and every sentence for language, grammar and punctuation. The aforementioned statements were also revised accordingly (see revised manuscript, lines 215-216 and lines 358 - 363).

OK

A reference is missing for the claims in lines 73-76.

We now support this claim with a relevant reference (see revised manuscript, lines 65 - 71 and reference #8).

OK